SCIENTIFIC CORRESPONDENCE

# Comment on 'Single nucleus sequencing reveals evidence of inter-nucleus recombination in arbuscular mycorrhizal fungi'

## Benjamin Auxier[1]*, Anna Bazzicalupo[2]

[1]Laboratory of Genetics, Wageningen University, Wageningen, Netherlands; [2]Department of Microbiology and Immunology, Montana State University, Bozeman, United States

**Abstract** Chen et al. recently reported evidence for inter-nucleus recombination in arbuscular mycorrhizal fungi (Chen et al., 2018a). Here, we report a reanalysis of their data. After filtering the data by excluding heterozygous sites in haploid nuclei, duplicated regions of the genome, and low-coverage depths base calls, we find the evidence for recombination to be very sparse.
DOI: https://doi.org/10.7554/eLife.47301.001

## Introduction

For many years arbuscular mycorrhizal fungi (AMF) were presumed to be asexual as no one had witnessed sexual structures in these fungi. This was puzzling because AMF retain core meiosis genes (*Halary et al., 2011*), indicating that a meiosis-like process most likely occurs in this lineage. Previous evidence for recombination later turned out to be based on duplicated gene copies (*Croll and Sanders, 2009*), or ribosomal RNA sequences that were paralogs (*Pawlowska and Taylor, 2004*; *Maeda et al., 2018*). Recently, based on work comparing single nuclei whole-genome sequences to bulk sequencing data, new evidence for recombination in these fungi was reported (*Chen et al., 2018a*). The isolates were dikaryotic, containing nuclei of two classes defined by their mating type (MAT) locus (*Ropars et al., 2016*). For each sequenced nucleus, PCR-amplification was attempted to assign a mating type class (MAT-1 up to MAT-5). Recombination was then inferred based on: (i) base-pair calls classed as one mating type found in the alternate mating type; (ii) nuclei of the same mating type showing variation in consecutive blocks of single nucleotide polymorphisms (SNPs); (iii) SNPs from nucleus 7 (SL1 strain) being more similar to SNPs of the alternate mating type, consistent with a recombination event spanning the MAT locus.

Here, we ask how strong the signal of within-strain recombination was in the data from *Chen et al. (2018a)* if we excluded heterozygous sites in haploid nuclei, duplicated regions of the genome, and low-coverage depths base calls. By removing data that cannot confidently be distinguished from sequencing errors and repeated regions, we find that the evidence for recombination is very sparse. We also report specific examples of these possible errors to justify our more stringent filtering of the data.

## Results

### The effect of filtering positions based on reads

Our first analysis was of the dataset reported in Supplementary file 6 of *Chen et al. (2018a)*, used for both Figures 2 and 3 of that manuscript. We filtered out: (1) positions where any single nucleus

*For correspondence:
ben.auxier@wur.nl

Competing interests: The authors declare that no competing interests exist.

was heterozygous (defined as sites with read depth >10, and alternate allele >10%), (2) any individual site with less than five reads coverage, and (3) positions with more than one high-confidence BLAST hit using the settings specified in *Chen et al. (2018a)*. We applied the filters individually or in combination, to see how many of the variable sites inferred as signals of recombination would be removed. Applying each filter individually removed between 19–77% of recombined positions. Filtering of low coverage sites had the strongest effect, a ~ 75% reduction. Applying all three filters together removed 91% of recombined sites (*Figure 1*). Notably, these filters had much less effect on the total number of analyzed sites, with the combined application of all three filters reducing the total number of sites by only ~22%.

Recombination, when involving crossing over, exchanges physical blocks between homologous chromosomes resulting in consecutive allelic differences. We calculated the number of recombined sites, as well as the number of recombined blocks, consecutive SNPs of the alternate haplotype, shown in *Table 1* (details of identified blocks can be found in *Table 1—source data 1*). Based on the analysis presented in Chen et al., all isolates show recombined blocks, in some cases spanning over a thousand base pairs. However, applying our three filters reduced these blocks. After filtering no consecutive SNPs remained for strain A4. Filtering applied to strains A5 and SL1 reduced the number of recombined blocks to 2 and 4, respectively, and also reduced the length of the remaining blocks.

## A specific example of repeated regions associated with biallelic sites in haploid nuclei

*Chen et al. (2018a)* interpreted consecutive SNPs differing between nuclei of the same mating type as a sign of recombination, as highlighted in Figure 3 of their article. The filtering used in *Chen et al. (2018a)* removed multi-copy sites '[i]f BLAST results returned more than two good hits', but retained regions with two BLAST hits. This could lead to the inclusion of SNPs that are heterozygous due to duplicated regions of the genome.

To show how repeated regions may lead to a false signal of recombination, we focused on an example highlighted in Figure 3 of *Chen et al. (2018a)*, and discussed in the main text of that article.

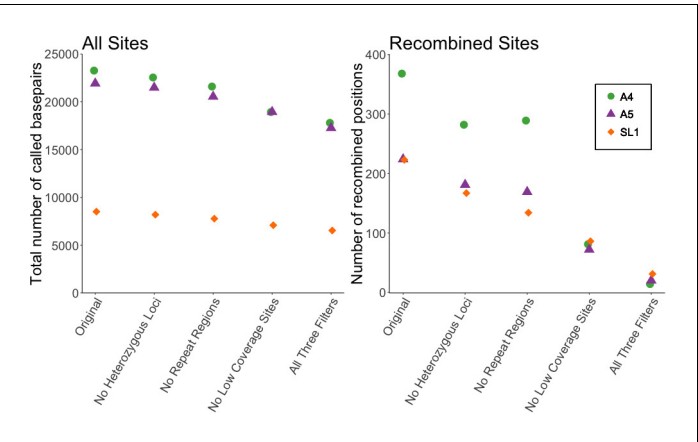

**Figure 1.** Filtering SNP data shows a decrease of 91% in number of recombined sites, but only ~20% decrease in total sites. Left panel shows the effect of filtering on all sites included in Supplementary file 6 of *Chen et al. (2018a)*, while right shows the effect on the number of recombined positions. Recombined positions identified based on second criterion in *Figure 1—figure supplement 1*. Different symbols show the effect on the three different strains (A4, A5, and SL1) used in our re-analysis.
DOI: https://doi.org/10.7554/eLife.47301.002
The following figure supplement is available for figure 1:

**Figure supplement 1.** Example of the two criteria used to identify recombination.
DOI: https://doi.org/10.7554/eLife.47301.003

**Table 1.** Lengths of recombination events before and after additional filtering.

| Isolate (Mating types) | Original data | | | Filtered data | | |
|---|---|---|---|---|---|---|
| | Number of recombined positions* | Number of recombined blocks (>1 consecutive SNP recombined) | Number of SNPs of longest recombined block (length in bp) | Number of recombined positions* | Number of recombined blocks (>1 consecutive SNP recombined) | Number of SNPs of longest recombined block (length in bp) |
| A4 *Mat-1/Mat-2* | 54/314 | 33 | 5 (1131) | 0/14 | 0 | 1 (1) |
| A5 *Mat-3/Mat-6* | 41/183 | 18 | 22 (2145) | 2/18 | 2 | 6 (670) |
| SL1 *Mat-5/Mat-1* | 111/112 | 20 | 16 (1872) | 22/9 | 4 | 6 (429) |

*numbers before/after the slash separate the two mating types, listed in the leftmost column. Number calculated based on the criteria shown in **Figure 1—figure supplement 1A**. **Table 1—source data 1** contains a list of all the recombined blocks identified.

DOI: https://doi.org/10.7554/eLife.47301.004

The following source data is available for  Table 1:

**Source data 1.** Recombined block locations.

List of all recombined blocks identified, which was used for **Table 1**.

DOI: https://doi.org/10.7554/eLife.47301.005

We found that several positions on scaffold 70 from isolate A4 were heterozygous in several nuclei (**Figure 2**), although they are treated as homozygous in **Chen et al. (2018a)**. High sequencing depth (>30) eliminates rare sequencing errors as the cause. To test if duplicated regions could be the cause of the heterozygosity, we performed a BLAST search against the A4 reference genome with sequences from scaffold 70:100354–100657. This search resulted in two BLAST matches: the self match on scaffold 70, as well as an additional match on scaffold 3570 (**Figure 2B**). When the short reads of the dikaryon (bulk sequencing of all nuclei) were aligned to the reference genome, both these SNPs on scaffold 70 and their match on scaffold 3570 were heterozygous, and the BLAST hit result for both showed 100% identity match. Thus, this repeated sequence seems to have been assembled as a chimera of the two variants in both scaffolds, and the short reads from either copy are mapped equally to both.

We feel this example illustrates the need to exclude repetitive regions from analyses of recombination.

## Confidence in low coverage sites to infer recombination

The data presented in **Chen et al. (2018a)** was filtered with a minimum of two reads. This is a very low threshold, and insufficient even for a consensus in the event of disagreeing reads. To look at the effect of low coverage on the signal of recombination, we compared the distribution of read depths between random and recombined sites. We first needed to identify recombined sites, as the method was lacking from the original manuscript, so we applied a parsimony criterion as detailed in **Figure 1—figure supplement 1**. While imperfect, this method certainly underestimates recombination, as it cannot identify recombined sites when equal numbers of nuclei within a mating type have alternate genotypes. Our method identified 733 positions, sufficient for analysis. Looking at the distribution of read depths, overall SL1 nuclei had ~95% fewer high coverage sites (average of 97 sites > 10 read depth for nuclei from SL1 versus 2290 for A4 and 2441 for A5) compared to A4 and A5 (**Figure 3**). We note here, as described in Table 1 of **Chen et al. (2018a)**, that SL1 nuclei cover much less of the genome (14%) compared to A4 and A5 (53% and 42%, respectively). Another fact visible from **Figure 3** is that, for A4 and A5, recombined sites are overrepresented by sites with low depth compared to sub-sampled non-recombined sites (Wilcoxon ranked sum test A4; $p=3.2\times10^{-09}$, A5 $p=2.9\times10^{-6}$). We note that for nuclei from isolate SL1, fewer overall recombined sites can be identified since the decreased breadth of coverage reduces overlap between nuclei, making it difficult to say whether this pattern of excess low-coverage sites is also present (p=0.11).

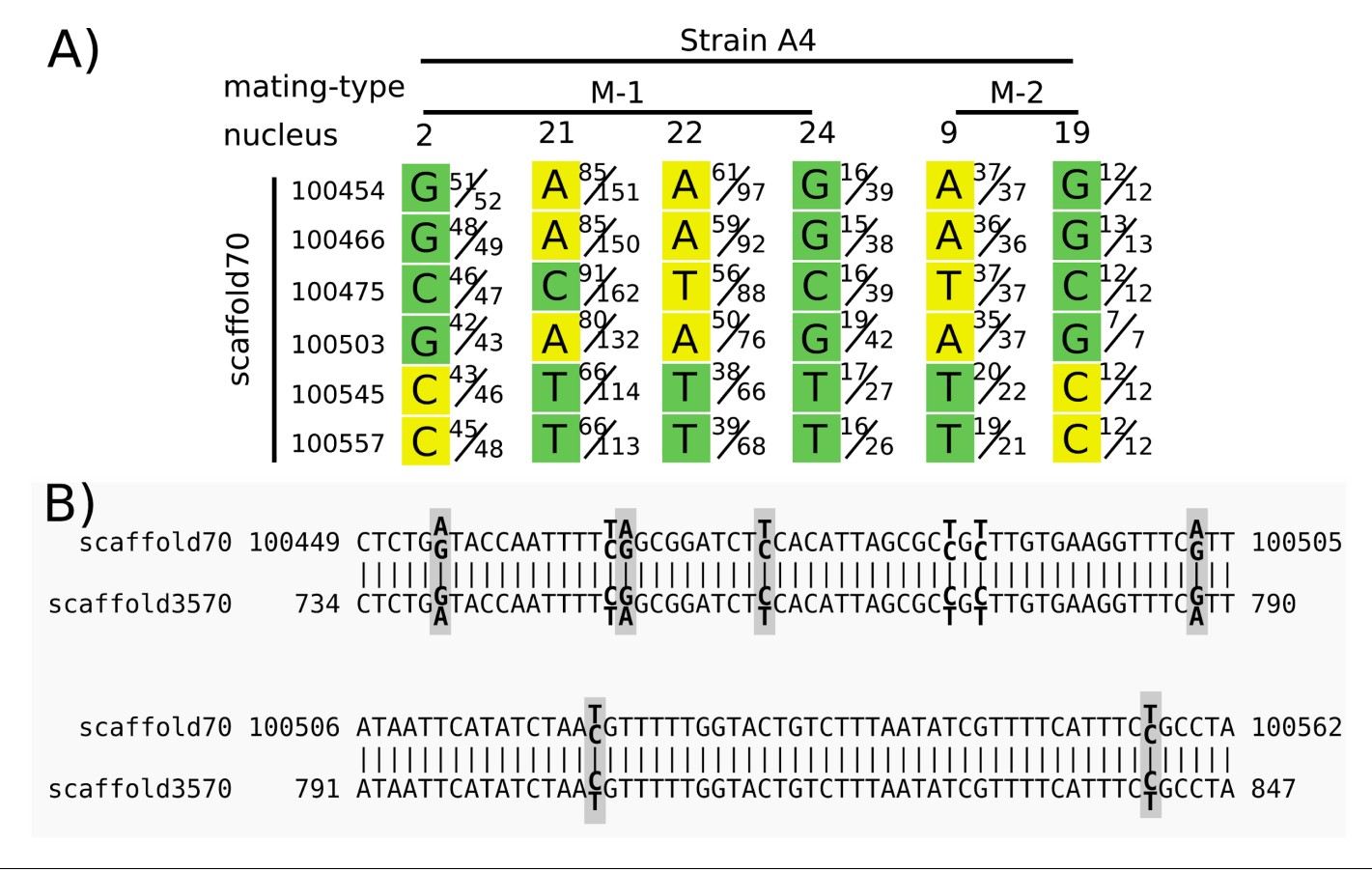

**Figure 2.** Heterozygous positions in single nuclei from a duplicated region were treated as homozygous in *Chen et al. (2018a)* and reported as evidence for a long stretch of recombination. Panel (**A**) shows the base calls for positions on scaffold 70 for six nuclei (four nuclei of mating type M-1 and two of M-2, as indicated in the top row). Each base call was assigned to a mating type class (green or yellow) in *Chen et al. (2018a)* based on an unspecified criterion. Variation between nuclei of the same mating type (e.g. variation among nuclei 2, 21, 22, and 24) is interpreted as recombination. We used their Illumina reads to show the ratio of reads supporting alternative nucleotides for each position. For example, in strain 4, mating-type 1, nucleus 2, position 100454, in Chen et al. the base was called as a G with a mating type 'green', and 51 of 52 reads matched G. However, for nucleus 21, only 85 of 151 reads supported an A at that position, while the other 66 supported a G. Panel (**B**) shows the alignment of the region shown in (**A**) with its best BLAST hit region on scaffold 3570. Heterozygous sites in the mapped reads of the dikaryon are indicated in bold, with the two alternate and reference bases shown slightly above/below. Gray boxes surround those sites included in (**A**). Note that both regions are heterozygous at the same aligned sites, and with the same alternate base for each heterozygous site. Graphic of (**A**) modified from Figure 3 of *Chen et al. (2018a)*, with the addition of nuclei 21 and 24.

DOI: https://doi.org/10.7554/eLife.47301.006

## Genome-wide pairwise SNP differences are reduced after filtering

We then assessed the evidence for genome-wide recombination based on pair-wise SNP differences. This is the analysis presented in Figure 3 of *Chen et al. (2018a)*, showing overall more recombination in SL1 than A4 and A5, indicated by a 'mosaic pattern'. We again note here that sequence from SL1 nuclei covers very little of the assembly (average of 14% from Table 1 of *Chen et al., 2018a*) means that very few positions will be covered between any two nuclei between nuclei of SL1 (14% in one nucleus x 14% in the other nucleus = 2% in both). After applying our filters, we find that in A4 and A5 almost all differences between nuclei of one mating type disappear (*Figure 4*). For nuclei from SL1, the filtering reduced the differences within a mating type, but since these nuclei cover so little of the genome, the overall dataset is reduced such that on average nuclei only share 9 SNPs that can be compared. Many nuclei have no overlapping SNPs and no comparison can be made (black squares in *Figure 4*). A few nuclei of opposite mating types, such as nuclei 17 and 25, show high similarity, but for these pairs the similarity is based on only one or two shared SNP positions.

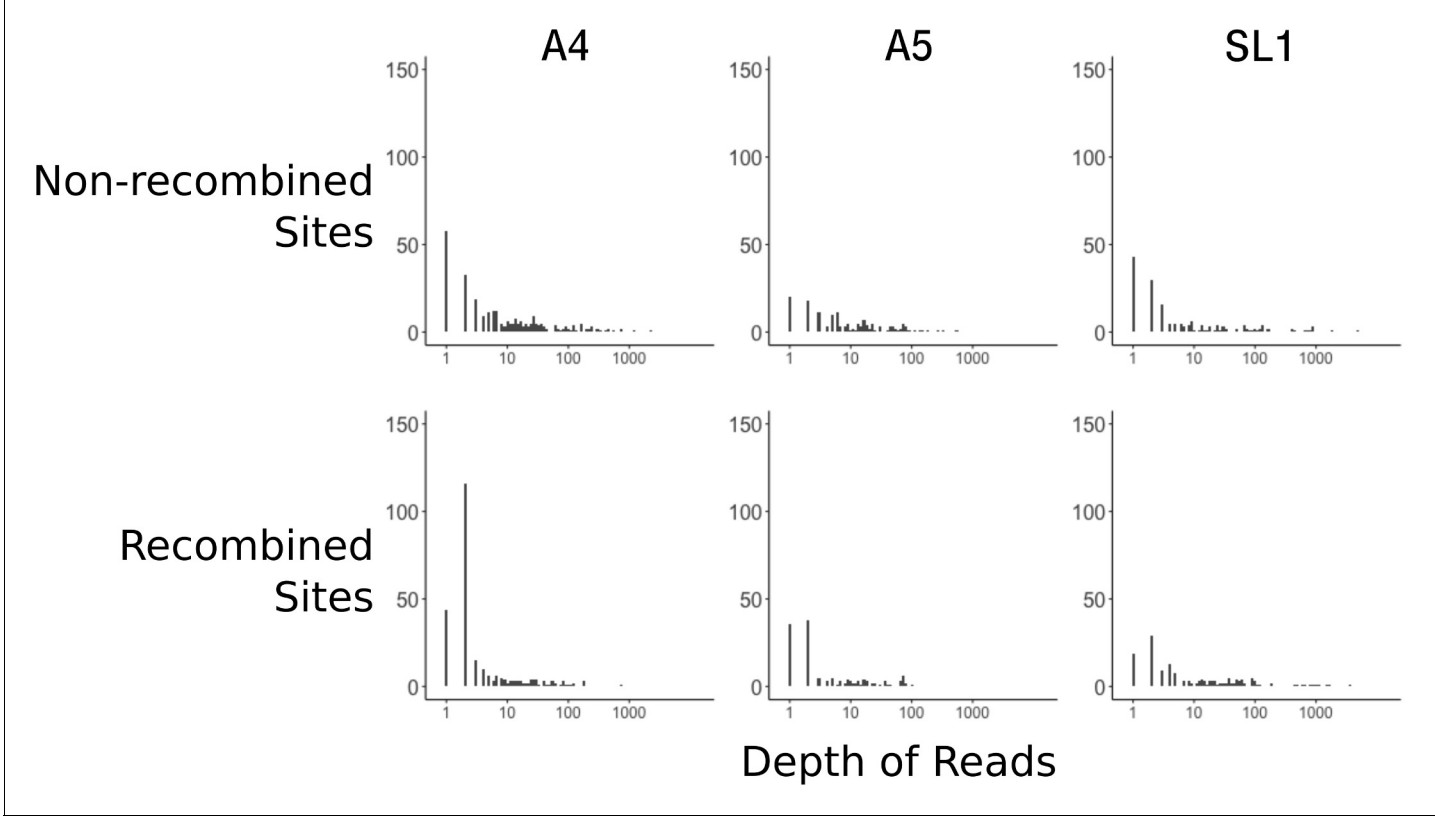

**Figure 3.** Recombined sites are overrepresented for low coverage sites. Top row: Distribution of sub-sampled read depths for non-recombined sites of individual nuclei for the three isolates, showing decreased coverage overall for SL1. Bottom row: Distribution of read depths for recombined sites does not mirror the distribution of random sites. Sites identified as recombined based on the parsimony criterion diagrammed in *Figure 1—figure supplement 1*. Note that read depth is plotted on a log scale.
DOI: https://doi.org/10.7554/eLife.47301.007

### Confirming the placement of nucleus 07 (SL1) could be strong evidence for recombination

In Figure 2 of *Chen et al. (2018a)*, nucleus 07 of SL1 shows a strong similarity with nuclei of the alternate mating type, seen by the clustering of nucleus 07 with *MAT-5*. However, this nucleus was PCR-genotyped to be of mating type *MAT*-1. The incongruence between PCR-genotyped mating type and the SNP clustering would be evidence of recombination spanning the *MAT* locus. To confirm this, we looked in the mapped reads of each nucleus to find reads mapping to either alternate mating type locus (*Figure 4—figure supplement 1*]). We found that there were no Illumina sequencing reads of nucleus 07 mapped to either mating type locus, indicating the whole genome amplification step may not have amplified the mating locus. As such, we have no available evidence for the mating type of this nucleus. Without corresponding Illumina evidence, we consider the PCR product the sole remaining evidence of this recombination event. This PCR experiment that is not confirmed with Illumina data represents the only remaining evidence for recombination after read filtering. We find cross-contamination of the PCR to be a more likely scenario in the face of many billions of sequenced bases from an Illumina run.

### Conclusion

Finding a balance between filtering poor data and losing informative data is a critical component of any analysis. For this dataset, we provide evidence for the necessity of stringent filtering to avoid inferences based on erroneous or misleading data. We do not consider our filters to be particularly strict, as removing low coverage sites, repeated regions, and heterozygous sites from haploid data is commonplace in genomic analyses. In SL1, three blocks of consecutive SNPs remained after our

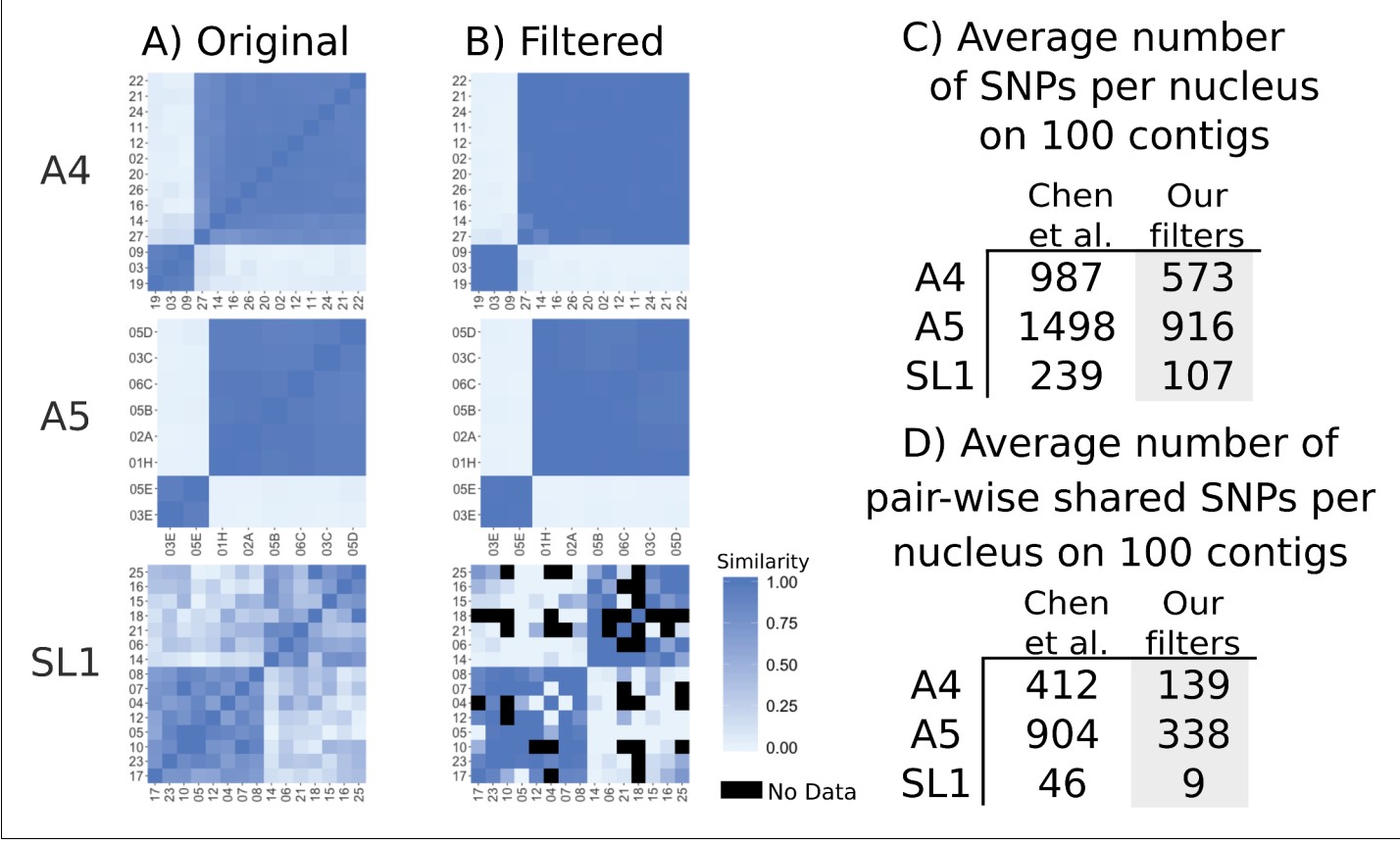

**Figure 4.** Genetic similarity of SL1 is strongly affected by SNP filtering methods. (**A**) left column panels show heatmaps generated from original data presented in Supplementary file 6 of Chen et al. (**B**) Right panels show data after filtering. Black squares represent pairs that do not share any SNPs. (**C**) Average number of SNPs in the dataset from *Chen et al. (2018a)* and after filtering. Note that SL1 has the lowest number of SNPs due to low sequencing breadth. (**D**) Average number of pairwise overlapping SNPs per nucleus, note that after filtering nuclei from SL1 have fewer than 10 SNPs overlapping on average, again due to low sequencing breadth.

DOI: https://doi.org/10.7554/eLife.47301.008

The following source data and figure supplements are available for figure 4:

**Figure supplement 1.** The critical mating type of nucleus 07 (black star) from SL1 is unsupported by short-read data.

DOI: https://doi.org/10.7554/eLife.47301.009

**Figure supplement 1—source data 1.** Mating loci identified in A4, A5, and SL1 genome assemblies.

DOI: https://doi.org/10.7554/eLife.47301.010

filtering, and two regions in A5. Some of these regions likely remain because our heterozygosity filter requires a minimum of ten reads, thus low coverage heterozygous sites are not excluded. This analysis used the first 100 contigs, covering approximately 10 Mb. As such, finding only a handful of putative recombined SNPs certainly cannot be confidently separated from amplification/sequencing noise. While small blocks of genetic exchange may be compatible with gene conversion, the limited number of markers involved greatly increases the difficulty in identifying high-confidence gene conversion events (*Wijnker et al., 2013*; *Qi et al., 2014*).

Mapping recombination inside repetitive regions would require longer reads than available from standard short-read technologies. This is a formidable task due to the input requirements of PacBio technologies, but it has been accomplished using linked short reads from individual pollen cells (*Sun et al., 2019*). Notably, the use of a more contiguous reference genome will actually include additional repetitive regions, and the exclusion of repetitive regions will become even more important. As the apparent recombined blocks are much smaller than the contigs, a more contiguous genome assembly would not change our analysis. Single nucleus genome amplification with multiple displacement amplification produces extremely variable genome coverage. Normalization of the

data was shown to improve the quality of AMF genomic data when coverage is variable (*Montoliu-Nerin et al., 2019*). In addition to normalization, low coverage sites could still be used with a model-based approach, which incorporates the associated uncertainty (*Hinch et al., 2019*; *Bloom et al., 2013*). Finally, removing heterozygous sites from haploid single-nuclei seems like a self-evident requirement.

The conserved meiosis genes found in the genomes of Glomeromycotan species strongly suggests a meiosis-like process allowing recombination and re-shuffling of genetic material among genomes. Uncovering the details of this process would be a major scientific breakthrough. Given this importance, claims made regarding recombination in Glomeromycota require rigorous examination. While we acknowledge that the models presented in *Chen et al. (2018a)* are valuable framework to test hypotheses for meiosis-like mechanisms found in these fungi, the data presented are not robust enough to support or reject them. As such, we believe that one of the greatest remaining mysteries in mycology remains unknown.

## Materials and methods

### Raw data
We obtained the SPAdes assemblies for the three dikaryotic *R. irregularis* isolates from NCBI, as well as the paired-end read libraries from the dikaryons and the paired-end reads of the single nuclei. Details of the accession numbers used are found in *Chen et al. (2018a)*.

### Read processing
Short reads were cleaned with FASTP (*Chen et al., 2018b*), then aligned using BWA mem as in *Chen et al. (2018a)*. Reads per nucleus were analyzed using the python modules pysam (*Heger and Jacobs, 2019*, and BLAST searches were scripted using biopython (*Cock et al., 2009*). Scripts used are available on GitHub repository (https://github.com/BenAuxier/Chen.2018.Response; *Auxier, 2019*; copy archived at https://github.com/elifesciences-publications/Chen.2018.Response). The parsimony criterion shown in *Figure 1—figure supplement 1A*. for identification of recombined sites, was performed manually. Filtering of excel files and calculations of recombined sites was performed with the R statistical language, which was also used to prepare plots with ggplot2 and distance matrices using ape (*Wickham, 2016*; *Paradis and Schliep, 2019*).

We compared the coverage representation between non-recombined and recombined sites. We subsampled from non-recombined sites to match the number of reads in recombined sites and performed Wilcoxon ranked sum test in R between the subsampled set of non-recombined reads and the recombined reads.

### Mating-type loci identification and mapping
As the location of the mating type loci was not specified in the results of *Chen et al. (2018a)*, we identified them based on data from *Ropars et al. (2016)*. Specifically, we used the primers sequences kary001, kary002, and kary003 as query sequences for BLAST searches against the A4, A5, and SL1 genomes. Each primer sequence had strong matches against two different scaffolds, consistent with divergent ideomorphs as found in *Ropars et al. (2016)*. As these primers only target a small region, we extended the locus using the annotations found on NCBI to identify the boundaries of the pair of genes. These locations are reported in *Figure 4—figure supplement 1—source data 1*.. As no annotations are available for SL1, we used the entire sequence of the ideomorph on scaffold511 of A5 as a BLAST query to identify the homologous regions.

With the mating locus identified, we then counted the number of reads that mapped to each sequence using samtools (*Li et al., 2009*).

### Calculation of overlapping SNPs
To calculate the expected number of overlapping SNPs found in *Figure 4D*, we used the following formula:

$$\left(\frac{\text{number of bases called for isolate in supplementary file 6}}{\text{number of positions}}\right)\left(\frac{1}{\text{number of nuclei}}\right)$$
$$= Proportion\ covered\ per\ nucleus$$

$$(Proportion\ covered\ per\ nucleus)^2 = Expected\ pairwise\ overlap\ proportion\ between\ nuclei$$

$$(Expected\ pairwise\ overlap\ proportion)(number\ of\ positions) = Expected\ number\ of\ shared\ positions$$

## Acknowledgements

We thank Dr. Duur Aanen, Dr. Anna Rosling, Dr. Marisol Sanchez-Garcia, Dr. Erik Wijnker, and Mathijs Nieuwenhuis for critical feedback. We also thank the three anonymous reviewers.

## Additional information

### Funding

The authors declare that there was no funding for this work.

### Author contributions

Benjamin Auxier, Conceptualization, Data curation, Formal analysis, Visualization, Writing—original draft, Writing—review and editing; Anna Bazzicalupo, Writing—original draft, Writing—review and editing

### Author ORCIDs

Benjamin Auxier https://orcid.org/0000-0002-7743-0610
Anna Bazzicalupo https://orcid.org/0000-0001-5845-9517

### Decision letter and Author response

Decision letter https://doi.org/10.7554/eLife.47301.014
Author response https://doi.org/10.7554/eLife.47301.015

## Additional files

### Supplementary files

• Transparent reporting form DOI: https://doi.org/10.7554/eLife.47301.011

### Data availability

No data was generated for this study. Analyses and scripts can be found on https://github.com/BenAuxier/Chen.2018.Response (copy archived at https://github.com/elifesciences-publications/Chen.2018.Response).

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
