## [Decision Letter]

Thank you for submitting your article "Comment on 'Single nucleus sequencing reveals evidence of inter-nucleus recombination in arbuscular mycorrhizal fungi'" for consideration by *eLife*. Your article has been reviewed by three peer reviewers, and the evaluation has been overseen by Raphael Mercier as the Reviewing Editor, Patricia Wittkopp as the Senior Editor, and Peter Rodgers, the *eLife* Features Editor. The reviewers have opted to remain anonymous.

We invite you to submit a revised version of your manuscript that addresses the comments of reviewer #2 and reviewer #3 (please see below; reviewer #1 did not raise any points for you to address). Please also submit a point-by-point response to these comments.

Reviewer #2:

The authors present a rather convincing case for a re-appraisal of the data and conclusions of Chen et al., 2018. However, there are two major issues that should be addressed before publication.

1) More information is needed in order for readers to be able to correctly interpret the data presented in Table 1 and to relate it to the parsimony methodology presented in Figure 1—figure supplement 1. Is the number of recombined positions as defined in Figure 1—figure supplement 1B? If so, what do the numbers before and after the slash in the table entry represent? The number of recombined positions out of the total number of positions? If so, then is the total number of positions counted as only those positions where all nuclei were able to be genotyped at that position? If that is the case, then why are these numbers different from the number of shared SNPs presented in Figure 4C? And why would the number before the slash be greater than the number after the slash (as it is for SL1 after filtering)? For "recombined blocks", does this refer to what is shown and described as "sites" in Figure 1—figure supplement 1A? More information is needed to describe the criteria used for this analysis. I can imagine three possible haplotype situations that could be considered as recombinant sites according to what is presented in Figure 1—figure supplement 1A: (a) all contiguous SNPs match the haplotype assigned to the opposite mating type, (b) only one of the contiguous SNPs matches the other haplotype, and c) multiple SNPs match the other haplotype.

These scenarios all have different likelihoods of representing true recombinant events. For example, scenario (a) only represents a true recombination event if the mating type is correctly assigned (as later discussed by the authors). For scenario (b), it is rather difficult to assign this to a recombination event, especially if it is only one SNP in the middle of a series of contiguous SNPs that would otherwise match the haplotype associated with its mating type or just a single SNP on that scaffold. Here, the most probable explanation might be a mutation. Scenario (c) has the strongest likelihood of representing a true recombination event, but only if there is a switch where there are contiguous SNPs from one haplotype to contiguous SNPs from another haplotype. Multiple haplotype switching would require multiple recombination events, which seems unlikely given that double crossovers are unlikely to occur within the short sequence lengths of the reported blocks in Supplementary file 2. It is not clear to me which of these scenarios is included in the parsimony criterion for sites. For positions, the attribution of SNP differences between mating types to recombination over other processes is unclear. In Figure 1—figure supplement 1B, for positions 5 and 6, only one nucleus is (D) is different from the others. This could arise through recombination or through mutation, (especially if it is only one SNP among many on a scaffold – which is not diagrammed in the example shown in Figure 1—figure supplement 1B, but could have been scored in the data). If it is several contiguous positions, then recombination is more likely than several independent mutations that would give rise to the other haplotype. The authors should clarify this in the text and/or by revising Figure 1—figure supplement 1.

2) My other major concern is the analysis presented in Figure 3 and associated commentary. The authors state "Another fact visible from Figure 3 is that for A4 and A5, recombined sites are overrepresented by sites with a read depth of 2 compared to random sites". I agree that it looks that way in the figure, but I would like to see some statistics to support this claim. The number of random sites is much, much greater than the number of recombined sites, further complicating a simple visual comparison. I suggest that the authors subsample the number of random sites to match the number of recombined sites and display this in the plot. Also, would "non-recombined" sites be a more appropriate term than random sites? The legend of this figure says that recombined sites are identified as recombined based on the parsimony criterion diagrammed in Figure 1—figure supplement 1. Is the read depth plotted for each individual SNP in a recombined site? Or is the average depth across all SNPs in a recombined site? The discrete values suggest that it is the former or that is actually referring to recombinant positions rather than sites.

Reviewer #3:

Auxier and Bazzicalupo express their concerns regarding an earlier study on inter-nucleus recombination in arbuscular mycorrhizal fungi, and tested their concerns regarding the data analysis by filtering the presented evidence for recombination using information on repetitive regions/heterozygous sites as well as short read alignment coverage information. In general, the search for rare events in the genome cannot be performed with relaxed filters, as false positives signal (even it is low) will affect rare events much more than frequent events.

Figure 1 is absolutely convincing and clearly shows that the new filtering more strongly affects the sites that are annotated as recombined sites, as compared to the sites that are not recombined. This technical difference between recombined sites and non-recombined sites can only be true if these two sets of sites are technically different. There is no reason why recombination should lead to such a difference. This is true for the repeat analysis/heterozygous site analysis as well as for the low coverage analysis. In consequence, I do share the concerns regarding the finding of the original study.

In consequence, I fully agree with the last statement of the manuscript: even though the new filtering does not remove all signals that could be recombination-induced, the (very little) remaining signal for recombination "are not robust enough to support or reject" meiosis-like mechanisms.

Moreover, I would not even agree that the remaining evidence for recombination, the PCR based assessment of the mating type of nucleus 07 (SL1), is a strong evidence for recombination. It is a non-replicated experiment of a single event (as far as I understand) and thus does not meet the general criteria to support a conclusion of such large impact.

My only concern regarding the comment:

Even if stated in the original manuscript, I would not agree that longer conversion tracts support recombination events more than single marker conversions. Recombination does not necessarily exchange long tracts, for example gene conversion-like events could be very short (as for example shown in Arabidopsis where the majority of gene conversions is only supported by single markers). If gene conversion like mechanisms would act here, short tracts might be the expected pattern. Therefore, I do not see why the absence of long tracts should be prominently illustrated.

---

## [Author Response]

Reviewer #2:The authors present a rather convincing case for a re-appraisal of the data and conclusions of Chen et al., 2018. However, there are two major issues that should be addressed before publication.1) More information is needed in order for readers to be able to correctly interpret the data presented in Table 1 and to relate it to the parsimony methodology presented in Figure 1—figure supplement 1. Is the number of recombined positions as defined in Figure 1—figure supplement 1B? If so, what do the numbers before and after the slash in the table entry represent? The number of recombined positions out of the total number of positions? If so, then is the total number of positions counted as only those positions where all nuclei were able to be genotyped at that position? If that is the case, then why are these numbers different from the number of shared SNPs presented in Figure 4C? And why would the number before the slash be greater than the number after the slash (as it is for SL1 after filtering)? For "recombined blocks", does this refer to what is shown and described as "sites" in Figure 1—figure supplement 1A?

Apologies for the confusion, we forgot to add the associated text to the table legend. The number before the slash is for nuclei from one mating type, and after the slash is for nuclei of the opposite mating type, with the mating types corresponding to the mating types listed in the leftmost column. These numbers differ from Figure 4C as Table 1 only includes sites identified by the criteria outlined in Figure 1—figure supplement 1, while Figure 4C is for all SNPs, between all nuclei.

More information is needed to describe the criteria used for this analysis. I can imagine three possible haplotype situations that could be considered as recombinant sites according to what is presented in Figure 1—figure supplement 1A: (a) all contiguous SNPs match the haplotype assigned to the opposite mating type, (b) only one of the contiguous SNPs matches the other haplotype, and c) multiple SNPs match the other haplotype.

It is unclear what contiguous SNPs refer to in this context. The SNPs identified by Chen et al. are rarely contiguous, as there are many invariant bases between SNPs. To clarify, no cases were identified by Chen et al. where all the SNPs on a single contig were of the opposite mating type.

To hopefully clarify further, our criteria were used to per SNP identify either recombined sites in an individual nucleus, or positions where recombination is supposed to have occur, without determining which nuclei were recombinant.

We understand that these criteria a not intuitive, but the authors of Chen et al. were either unwilling or unable to provide the criteria that they used for Figure 3 of their manuscript, and the original manuscript does not explain how colors in Figure 3 were assigned. Any analysis of the underlying data requires an objective set of criteria, which Chen and co-authors failed to provide.

These scenarios all have different likelihoods of representing true recombinant events. For example, scenario (a) only represents a true recombination event if the mating type is correctly assigned (as later discussed by the authors). For scenario (b), it is rather difficult to assign this to a recombination event, especially if it is only one SNP in the middle of a series of contiguous SNPs that would otherwise match the haplotype associated with its mating type or just a single SNP on that scaffold. Here, the most probable explanation might be a mutation. Scenario (c) has the strongest likelihood of representing a true recombination event, but only if there is a switch where there are contiguous SNPs from one haplotype to contiguous SNPs from another haplotype. Multiple haplotype switching would require multiple recombination events, which seems unlikely given that double crossovers are unlikely to occur within the short sequence lengths of the reported blocks in Supplementary file 2.

We agree with this logic, and we note that all of the claimed recombination events would involve double crossovers, as the haplotypes revert back to the original mating type in every case. Short stretches of this could indeed represent mutation, or alternatively gene conversion as emphasized by reviewer #3.

It is not clear to me which of these scenarios is included in the parsimony criterion for sites. For positions, the attribution of SNP differences between mating types to recombination over other processes is unclear. In Figure 1—figure supplement 1B, for positions 5 and 6, only one nucleus is (D) is different from the others. This could arise through recombination or through mutation, (especially if it is only one SNP among many on a scaffold – which is not diagrammed in the example shown in Figure 1—figure supplement 1B, but could have been scored in the data). If it is several contiguous positions, then recombination is more likely than several independent mutations that would give rise to the other haplotype. The authors should clarify this in the text and/or by revising Figure 1—figure supplement 1.

We agree with the reviewer and have tried to clarify our criteria linguistically. Additionally, we have modified Figure 1—figure supplement 1A to represent how we interpret singletons. We point to Figure 3 of Chen et al.’s original manuscript, where they use examples of singletons as well as contiguous regions.

2) My other major concern is the analysis presented in Figure 3 and associated commentary. The authors state "Another fact visible from Figure 3 is that for A4 and A5, recombined sites are overrepresented by sites with a read depth of 2 compared to random sites". I agree that it looks that way in the figure, but I would like to see some statistics to support this claim. The number of random sites is much, much greater than the number of recombined sites, further complicating a simple visual comparison. I suggest that the authors subsample the number of random sites to match the number of recombined sites and display this in the plot.

We tested the conclusion of Figure 2 using the Wilcoxon ranked sum test. The statistical results are consistent with our interpretation.

Results:

“Another fact visible from Figure 3 is that, for A4 and A5, recombined sites are overrepresented by sites with low depth compared to non-recombined sites (Wilcoxon ranked sum test A4; p=3.2x10^-09^, A5 p=2.9x10^-6^). We note that for nuclei from isolate SL1, fewer overall recombined sites can be identified since the decreased breadth of coverage reduces overlap between nuclei, making it difficult to say whether this pattern of excess low-coverage sites is also present (p=0.11).”

Materials and methods:

“We compared the coverage representation between non-recombined and recombined sites. We subsampled from non-recombined sites to match the number of reads in recombined sites and performed Wilcoxon ranked sum test in R between the subsampled set of non-recombined reads and the recombined reads.”

Also, would "non-recombined" sites be a more appropriate term than random sites?

The original non-subsampled analysis did not exclude identified recombined sites, but we have excluded recombined sites from the subsampled analysis. We have changed the wording of Figure 3 accordingly.

The legend of this figure says that recombined sites are identified as recombined based on the parsimony criterion diagrammed in Figure 1—figure supplement 1. Is the read depth plotted for each individual SNP in a recombined site? Or is the average depth across all SNPs in a recombined site? The discrete values suggest that it is the former or that is actually referring to recombinant positions rather than sites.

Yes, it is the read depth per nucleus. We have clarified the legend of the figure.

Reviewer #3:Auxier and Bazzicalupo express their concerns regarding an earlier study on inter-nucleus recombination in arbuscular mycorrhizal fungi, and tested their concerns regarding the data analysis by filtering the presented evidence for recombination using information on repetitive regions/heterozygous sites as well as short read alignment coverage information. In general, the search for rare events in the genome cannot be performed with relaxed filters, as false positives signal (even it is low) will affect rare events much more than frequent events.Figure 1 is absolutely convincing and clearly shows that the new filtering more strongly affects the sites that are annotated as recombined sites, as compared to the sites that are not recombined. This technical difference between recombined sites and non-recombined sites can only be true if these two sets of sites are technically different. There is no reason why recombination should lead to such a difference. This is true for the repeat analysis/heterozygous site analysis as well as for the low coverage analysis. In consequence, I do share the concerns regarding the finding of the original study.In consequence, I fully agree with the last statement of the manuscript: even though the new filtering does not remove all signals that could be recombination-induced, the (very little) remaining signal for recombination "are not robust enough to support or reject" meiosis-like mechanisms.Moreover, I would not even agree that the remaining evidence for recombination, the PCR based assessment of the mating type of nucleus 07 (SL1), is a strong evidence for recombination. It is a non-replicated experiment of a single event (as far as I understand) and thus does not meet the general criteria to support a conclusion of such large impact.

We agree, but without any direct evidence against we feel the need to give the authors the benefit of the doubt.

My only concern regarding the comment:Even if stated in the original manuscript, I would not agree that longer conversion tracts support recombination events more than single marker conversions. Recombination does not necessarily exchange long tracts, for example gene conversion-like events could be very short (as for example shown in Arabidopsis where the majority of gene conversions is only supported by single markers). If gene conversion like mechanisms would act here, short tracts might be the expected pattern. Therefore, I do not see why the absence of long tracts should be prominently illustrated.

We discuss the length of tracts because they are highlighted by Chen et al. in their Results section:

“In many cases, recombining genotypes encompass hundreds to thousands of base pairs, (Figure 3, Supplementary file 6). […] In this example, a single recombination event between genotypes harbored by the nuclei 22 (*MAT*-1) and 19 (*MAT*-2) resulted in a genetic exchange involving at least one thousand base pairs, and similar events are found elsewhere in the genome of A4.”

While we agree that gene conversion could result in the transfer of single markers, Chen et al. refer to “meiotic-like processes” in the Abstract of their publication. If indeed a meiotic-like process is occuring then gene conversion should be paired with at least one crossover event per chromosome for proper segregation. It is possible that crossover events are only occurring on the distal ends of chromosomes, as in *Agaricus bisporus*, and these distal ends are not included in the largest 100 scaffolds. But there is no evidence presented for this scenario and we prefer not to speculate on potential mechanisms to explain Chen et al.’s low quality data.

In consideration of the reviewer’s and editors comments, we have added a sentence to the conclusion acknowledging that gene conversion events would be of a size consistent with the size found by Chen et al., however gene conversion events are extremely difficult to confidently identify, as shown by Qi et al. who we have added as a reference as well as Wijnker et al.

“While small blocks of genetic exchange may be compatible with gene conversion, the limited number of markers involved greatly increases the difficulty in identifying high-confidence gene conversion events (Wijnker et al., 2013; Qi et al., 2014).”